# Single Nucleotide Polymorphisms Associated with Metformin and Sulphonylureas’ Glycaemic Response among South African Adults with Type 2 Diabetes Mellitus

**DOI:** 10.3390/jpm11020104

**Published:** 2021-02-06

**Authors:** Charity Masilela, Brendon Pearce, Joven Jebio Ongole, Oladele Vincent Adeniyi, Mongi Benjeddou

**Affiliations:** 1Department of Biotechnology, University of the Western Cape, Bellville 7535, South Africa; brendon.biff@gmail.com (B.P.); mbenjeddou@uwc.ac.za (M.B.); 2Center for Teaching and Learning, Department of Family Medicine, Piet Retief Hospital, Mkhondo 2380, South Africa; jjongole@gmail.com; 3Department of Family Medicine, Walter Sisulu University, East London 5200, South Africa; vincoladele@gmail.com

**Keywords:** type 2 diabetes, single nucleotide polymorphisms, metformin, sulphonylureas, combination therapy

## Abstract

Aims: To examine the association of polymorphisms belonging to *SLC22A1*, *SP1*, *PRPF31*, *NBEA*, *SCNN1B*, *CPA6* and *CAPN10* genes with glycaemic response to metformin and sulphonylureas (SU) combination therapy among South African adults with diabetes mellitus type 2 (T2DM). Methods: A total of 128 individuals of Swati (*n* = 22) and Zulu (*n* = 106) origin attending chronic care for T2DM were recruited. Nine SNPs previously associated with metformin and SUs were selected and genotyped using MassArray. Uncontrolled T2DM was defined as HbA1c > 7%. The association between genotypes, alleles and glycaemic response to treatment was determined using multivariate logistic regression model analysis. Results: About 85.93% (*n* = 110) of the study participants were female and 77.34% (*n* = 99) had uncontrolled T2DM (HbA1c > 7%). In the multivariate (adjusted) logistic regression model analysis, the CC genotype of rs2162145 (*CPA6*), GG and GA genotypes of rs889299 (*SCNN1B*) were significantly associated with uncontrolled T2DM. On the other hand, the C allele of rs254271 (*PRPF31*) and the GA genotype of rs3792269 (*CAPN10*) were associated with controlled T2DM. A significant interaction between rs2162145 and rs889299 in response to metformin and SU combination therapy was observed. Conclusions: In this study, we reported the association of rs2162145 (CC) and rs889299 (GG and GA) with uncontrolled T2DM. We also reported the association of rs254271 (C) and rs3792269 (GA) with controlled T2DM in response to metformin and SU combination therapy. Furthermore, an interaction between rs2162145 and rs889299 was established, where the genotype combination GA (rs889299) and TT (rs2162145) was associated with uncontrolled T2DM.

## 1. Introduction

Diabetes mellitus (DM) is a chronic non-communicable disease that affects about 463 million people worldwide [1]. By the year 2030, the burden of DM is expected to increase by 10.2%, reaching an estimated 578 million cases globally [1]. According to the International Diabetes Federation (IDF), over 4.5 million South African adults were estimated to be living with DM in the year 2019. Furthermore, over two million of these individuals were undiagnosed and were at a higher risk for life-threatening complications associated with DM [1]. Using data from the South African National Health and Nutrition Survey (2011–2012), Stokes et al. showed that 18.1% of South Africans were treated but exhibited poor glycaemic control (HbA1C > 7%) [2]. Uncontrolled DM and its complications have a huge and rapidly growing impact on the South African health care system [1,2,3]. Thus, improving glycaemic control in individuals diagnosed with DM and initiated on treatment will require concerted efforts from many fronts including individual behavioural changes, public health efforts and tailored medical care that is guided by pharmacogenomics strategies.

Diabetes mellitus, particularly type 2 diabetes (T2DM), is a complex metabolic disease that is characterised by hyperglycaemia as a consequence of defects in insulin secretion, insulin action or a combination of both [4]. To date, there are more than ten classes of drugs that are used to manage T2DM [5,6]. However, metformin is the only approved anti-diabetic drug under the class of biguanides that is indicated for the treatment of T2DM [7]. Metformin exerts its anti-diabetic properties by decreasing hepatic glucose production, decreasing intestinal absorption of glucose and increasing insulin sensitivity. Thus, improving glucose uptake and utilisation in peripheral tissues [8,9]. On these grounds, metformin is the preferred initial oral anti-diabetic agent [10]. However, when monotherapy fails to achieve glycaemic goals, combination therapy using a second agent with a different mechanism of action is often initiated [6,11,12]. Sulphonylureas (SUs) are characterised as insulin secretagogues, as they stimulate insulin secretion in the pancreatic beta-cells. This class of drugs may also improve peripheral and hepatic insulin sensitivity by reducing glucose toxicity [12,13,14]. The most commonly prescribed SUs are glibenclamide, gliclazide and glimepiride [12,15,16]. These three drugs are classified as second-generation SUs and they are the preferred add-on agents to metformin therapy [6]. Despite these efforts, treatment with any class of anti-diabetic drug features variability that is brought by single nucleotide polymorphisms (SNPs) in genes that are directly or indirectly implicated in the pharmacokinetics and pharmacodynamics of anti-diabetic agents [9].

Specificity protein 1 (SP1) is a zinc finger transcription factor that binds to GC-rich motifs of many promoters including those found in the solute carrier gene superfamily that is responsible for metformin transport [17]. Calpain 10 (CAPN10) is calcium-dependent cysteine proteases that is implicated in glucose metabolism and pancreatic beta-cell function [18]. Owing to their respective roles in metformin transport and glucose metabolism, SNPs in genes coding both proteins were associated with insulin resistance, T2DM and response to anti-diabetic drugs [19,20]. For instance, the G allele of rs3792269 (*CAPN10*) was associated with an absolute reduction of HbA1c following a six-month treatment with metformin among Caucasian patients with T2DM [21]. On the other hand, rs2683511 (*SP1*) was associated with decreased metformin secretory clearance among a mixed American cohort, however, this effect was demonstrated among healthy individuals [22].

Carboxypeptidase A6 (CPA6) is an enzyme that is encoded by the *CPA6* gene. The enzyme is responsible for catalysing the release of C-terminal amino acids and have functions ranging from digestion to selective biosynthesis of neuroendocrine peptides [23]. On the other hand, pre-mRNA processing factor 31 (PRPF31) encodes a ubiquitously expressed mRNA splicing factor [24]. Furthermore, Rotroff et al. (2018) demonstrated that rs254271 (*PRPF31*) was associated with decreased metformin response among patients of European and African origin. Whilst rs2162145 of *CPA6* was associated with better response to metformin in the same study cohort, it was further demonstrated that rs57081354, an intronic polymorphism found in the Neurobeachin (*NBEA*) gene, was associated with a decreased metformin response [25]. While the role of these genes in the pharmacokinetics and pharmacodynamics of metformin is unknown, recently published data suggest that polymorphisms situated in these genes may predict metformin response in individuals with T2DM [25].

In addition, solute carrier family 22 member 1 (*SLC22A1*) is a poly-specific organic cation transporter encoded by the *SLC22A1* gene that plays an important role in the influx of metformin in hepatocytes and its elimination through the renal system [26]. A number of *SLC22A1* SNPs have been associated with variable metformin response in individuals with T2DM [27]. For instance, it has been shown that European carriers of the del/del genotype of rs36056065 (*SLC22A1*) may have a decreased but not absent risk of gastrointestinal side effects associated with metformin [28]. On the other hand, carriers of the del/del genotype of rs72552763 (*SLC22A1*) exhibited decreased hepatic distribution and exposure to metformin in healthy individuals residing in Denmark [26]. Moreover, Lebanese carriers of the AC and AA genotype of rs622342 (*SLC22A1*) showed a greater HbA1c reduction following metformin/SU combination therapy [29].

It was further demonstrated that rs889299 was associated with T2DM and that European carriers of the AA genotype may have an increased risk of oedema when treated with glibenclamide [30]. The rs889299 polymorphism occurs in Sodium Channel Epithelial 1 Subunit Beta (*SCNN1B*), a gene that encodes for the beta-subunit of epithelial sodium channel (ENaC) [30]. Literature suggest that ENaC activity may be regulated by ATP-binding cassette protein such as the K channel-associated sulfonylurea receptor [31]. Glibenclamide is a known inhibitor of the K channel-associated sulfonylurea receptor. Furthermore, the drug increased transepithelial Na transport in vitro [31]. On these grounds, it is possible that rs889299 may influence glycaemic response in individuals with T2DM undergoing SU monotherapy or metformin/SU combination therapy. Additionally, it is possible that epistatic interactions between this variant and co-existing polymorphism may influence glycaemic response to metformin/SU combination therapy. This effect is yet to be established in patients with T2DM of African descent.

It is becoming increasingly evident that a more personalised approach may be beneficial in the management of T2DM. Given the dearth of pharmacogenomics researches among individuals of African ancestry, data generated from other population groups are unlikely to reflect the overall effect of SNPs in anti-diabetic response among Africans. As such, there is an urgent need to investigate this identified gap with specific focus on polymorphisms that define response to metformin/SU combination therapy in African population. This study examines the association of nine polymorphisms belonging to *SLC22A1*, *SP1*, *PRPF31*, *NBEA*, *SCNN1B*, *CPA6* and *CAPN10* genes with glycaemic response to metformin/SU combination therapy among South African adults with T2DM. In addition, the study further assesses the epistatic interactions between these SNPs and their response to metformin/SU combination therapy.

## 2. Materials and Methods

### 2.1. Study Design and Patient Selection

Ethical clearance for this study was obtained from the Senate Research Committee of the University of the Western Cape (Ethics clearance number BM/16/5/19). Permission to implement the study was granted by the clinical governance of Piet Retief Hospital and the department of Health of the Mpumalanga Provinces. All participants received information on the purpose and procedure of the study in the home language (Swati and Zulu) prior to signing an informed consent by each participant.

A total of 128 individuals of Swati (*n* = 22) and Zulu (*n* = 106) origin attending chronic care for T2DM were recruited consecutively between January 2019 and June 2019, from the outpatient department of Piet Retief Hospital, Thandukukhanya Community Health Center and Mkhondo Town Clinic (Mkhondo, Mpumalanga). The study included participants who were 18 years or older and were on continuous Metformin/SU dual therapy for T2DM for at least a year prior to the study. Patients who were pregnant, diagnosed with Type 1 diabetes mellitus, malignancies, chronic kidney and liver disease, as well as those who were undergoing monotherapy of either insulin, metformin or any other drug for T2DM were excluded.

### 2.2. Data Collection

Anthropometric measurements were conducted by a trained research nurse. The weight of each participant was measured to the nearest 0.1 kg using a digital scale (Tanita-HD 309, Creative Health Products, MI, USA) and height to the nearest of 0.1 cm using a mounted stadiometer, with participants wearing minimal clothing. Body mass index (BMI) for each patient was estimated as weight (kg) divided by height (m^2^). We further categorised BMI as: underweight = BMI < 18.5kg/m^2^; normal weight = BMI: 18.5–24.9 kg/m^2^; overweight = BMI: 25.0–29.9 kg/m^2^; obese = BMI ≥ 30 kg/m^2^. Blood assays for glycated haemoglobin (HbA1c) were conducted by the National Health Laboratory Services (NHLS) in accordance with standardized protocols. Uncontrolled T2DM was defined as HbA1c > 7% in accordance with the guidelines of the Society for Endocrinology, Metabolism and Diabetes of South Africa.

Duration of T2DM and anti-diabetic drugs prescribed for each participant were retrieved from their clinical records. Prescribed SUs were glibenclamide, glimepiride and gliclazide in combination with metformin. Age, ethnicity and physical activity were self-reported and documented in a proforma designed for this study. Physical activity was classified into active if participants engaged in rigorous physical activity that increased heart rate, and inactive if participants did not take part in any form of physical activity. Ethnicity was defined as belonging to a social group with a common language, cultural tradition or ancestry; Swati or Zulu. DNA samples were collected from each participant in the form of buccal swabs and stored at −20 °C until they were extracted.

### 2.3. DNA Isolation

Genomic DNA was isolated from buccal swabs using a standard salt lysis method [32]. Extracted DNA samples were stored at −20 °C. DNA was quantified using a NanoDrop™2000/2000c UV/VIS Spectrophotometer (ThermoScientific™). SNPs were genotyped using the MassARRAY^®^System IPLEX extension reaction (Agena Bioscience™). Genotypes of the selected SNP variants were determined for all the study participants.

### 2.4. Selection of SNPs and Genotyping

Nine SNPs previously associated with Metformin or Sulfonylurea treatment outcome were selected using Pharmacogenomics Knowledge Base, Ensembl as well as an extensive survey of recent literature. Two multiplex MassARRAY systems (Agena Bioscience TM) were designed and optimized by Inqaba Biotechnical Industries (Pretoria, South Africa) in January 2017. Each multiplex was used to genotype selected SNPs, using an assay that is based on a locus-specific PCR reaction. This reaction is followed by a single base extension using the mass-modified dideoxynucleotide terminators of an oligonucleotide primer, which anneals upstream of the site of mutation. Matrix-assisted laser desorption/ionization—time-of-flight (MALDI-TOF) mass spectrometry was used to identify the SNP of interest.

### 2.5. Statistical Analysis

Statistical analyses were performed using IBM Statistical Package for Social Science (SPSS) Version 25 for Windows (IBM Corps, Armonk, New York, NY, USA). The general characteristics of the participants were expressed as frequency (percentages). The associations between alleles, genotypes and glycaemic response to metformin/SU combination therapy were assessed by multivariate logistic regression model analysis (unadjusted and adjusted odds ratios) and their 95% confidence intervals. The final model of the adjusted logistic regression analysis included rs2162145, rs2282143, rs254271, rs2683511, rs3792269, rs57081354, rs72552763, rs36056065 and rs622342. Results for the unadjusted logistic regression model analysis were expressed as crude odds ratios (CORs) and adjusted odds ratios (AORs) for the adjusted logistic regression model analysis. A *p*-value less than 0.05 was considered statistically significant. Bonferroni corrected *p*-values were set at <0.0125. The minor allele frequency (MAF) and Hardy–Weinberg equilibrium (HWE) tests were calculated using Genetic Analysis in Excel (GenAIEx) Version 6.5. SNP-SNP interactions between rs5708135, rs2162145, rs36056065, rs622342 and rs889299 were determined using Multi-factor dimensionality reduction (MDR) version 3.0.2. *Sp1* rs2683511 (TT), rs3792269 (GG) and rs72552763 (del/del) were not detected; therefore, they were excluded from the analysis. The best model of interaction was selected on the basis of a high cross-validation consistency (CVC) score and *p*-values. *p*-values were calculated using *x^2^* test, values <0.05 were deemed significant.

## 3. Results

### 3.1. General Characteristics of the Study Cohort

A total of 128 individuals with T2DM undergoing metformin/SU combination therapy were recruited. About 14.06% (*n* = 18) were male and 85.93% (*n* = 110) were female, of whom 35.93% (*n* = 46) were aged between 55 to 65 years. Furthermore, the cohort was comprised of 82.81% (*n* = 106) and 17.19% (*n* = 22) individuals of Zulu and Swati origin, respectively. Majority of the study participants (68.75%) were obese (BMI ≥ 30 kg/m^2^), 60.93% (*n* = 78) were inactive, 77.34% (*n* = 99) had uncontrolled T2DM (HbA1c > 7%) and 73.44% (*n* = 94) have been living with T2D for <5 years (Table 1).

### 3.2. Expression and Association of SNPs with Metformin/SU Combination Therapy Response

Seven (rs2162145, rs2282143, rs254271, rs2683511, rs3792269, rs57081354, rs72552763) out of nine SNPs were within the Hardy–Weinberg equilibrium (HWE) with *p*-values ranging from 0.134–0.771 (Table 2).

In the multivariate logistic regression (unadjusted) model analysis, the CC of rs622342 (COR = 4.65; 95% CI 1.08–19.86; *p* = 0.038), rs889299 (COR = 3.55; 95% CI 1.11–11.33; *p* = 0.032) and C allele of rs57081354 (COR = 2.15; 95% CI 1.14–4.04; *p* = 0.017) were significantly associated with uncontrolled DM, whilst the G allele of rs3792269 (COR = 0.33; 95% CI 0.12–0.88; *p* = 0.027) was associated with controlled DM (Table 3). No association was observed between uncontrolled DM and the genotypes or alleles of rs2162145, rs2683511, rs36056065, rs72552763 and rs254271 (*p* > 0.05).

After adjusting with each SNP, the multivariate logistic regression (adjusted) model analysis showed that the CC genotype rs2162145 (*CPA6)* (AOR = 14.86; 95% CI 1.71–29.04; *p* = 0.014), GG (AOR = 7.91; 95% CI 1.67–37.27; *p* = 0.009) and GA (AOR = 5.27; 95% CI 1.19–23.19; *p* = 0.028) genotypes of rs889299 (*SCNN1B*) were significantly associated with uncontrolled T2DM. On the other hand, the C allele of rs254271 (AOR = 0.20; 95% CI 0.07–0.62; *p* = 0.005) and the GA genotype of rs3792269 (*CAPN10*) (AOR = 0.15; 95% CI 0.02–0.85; *p* = 0.033) were associated with controlled T2DM (Table 3). Polymorphisms rs2683511, rs57081354, rs36056065, rs622342 and rs72552763 were not associated with uncontrolled DM in response to Metformin/SU combination therapy (*p* > 0.05). The Bonferroni correction *p*-value was set at < 0.0125. After Bonferroni correction, the rs3792269 (GA), rs254271 (C), rs2162145 (CC) and rs889299 (GG and GA) remained significant with *p*-values < 0.0125.

### 3.3. Epistatic Interaction Patterns between SNPs and Their Association with Response to Metformin/SU Combination Therapy

Epistatic interactions between *CPA6*, *PRPF31*, *SLC22A1*, *NBEA* and *SCNN1B* were analysed using Multifactor dimensionality reduction (MDR). The combination of rs21621459 (*CPA6*) and rs889299 (*SCNN1B*) demonstrated a high CVC score (6/10), and it was significantly associated with metformin/SU combination therapy outcome (*p* = 0.0022). The combination of rs2162145 (*CPA6*), rs622342 (*SLC22A1*) and rs889299 (*SCNN1B*) showed a low CVC score (5/10) (Table 4).

The genotype combinations GA (rs889299) and TT CPA6 rs2162145, GA (rs889299) and TC (rs2162145), and TT (rs2162145) and GG (rs889299) were prominently detected among patients with uncontrolled T2DM (HbA1c > 7%) The combination GA (rs889299) and TT rs2162145 was associated with uncontrolled T2DM (Figure 1). Other possible interactions between SNPs are demonstrated in Figure 2.

## 4. Discussion

The combination of metformin and SUs is among the most commonly prescribed dual therapies for the treatment of T2DM. Although widely prescribed, treatment outcome with oral anti-diabetic drugs differs strongly between individuals due to genetic factors. Accounting for these factors would lead to more personalised treatment regimens and help combat the increasing prevalence of uncontrolled T2DM. Therefore, the current study investigated the association of nine polymorphisms belonging to *SLC22A1*, *SP1*, *PRPF31*, *NBEA*, *SCNN1B*, *CPA6* and *CAPN10* genes with glycaemic response to metformin/SU combination therapy. The study further assessed genetic interactions between these SNPs and glycaemic response to metformin/SU combination therapy among South African adults with T2DM.

In this study, we investigated the effect of two *SLC22A1* polymorphisms (rs36056065 and rs622342) on glycaemic response to metformin/SU combination therapy in patients with T2DM. The CC genotype of rs622342 was significantly associated with uncontrolled T2DM. The *SLC22A1* gene plays a crucial role in metformin transport. As such, polymorphisms in this gene have been associated with metformin response among patients with T2DM [33]. In a South India population, Umamaheswaran et al. [33] demonstrated that carriers of allele C of rs622342 showed decreased response to metformin therapy. It was further demonstrated that this effect was more pronounced among carriers of two copies of the C allele [33]. Furthermore, Naja et al. [29] showed that Lebanese carriers of the AC or the AA genotype exhibited better glycaemic control in individuals with T2DM undergoing metformin/SU combination therapy. Similar effects were observed among Egyptian patients of T2DM [34]. These findings warrant the use of this polymorphism as a predictor of metformin/SU efficacy among patients of African origin with T2DM.

We investigated the effect of rs2162145 (*CPA6*), rs57081354 (*NBEA*) and rs254271 (*PRPF31*) on glycaemic response to metformin/SU combination. Our findings suggest that Swati and Zulu carriers of the CC genotype of rs2162145 and C allele of rs57081354 were more likely to exhibit uncontrolled T2DM in response to metformin/SU combination therapy. Whereas carriers of the minor allele C of rs254271 were more likely to exhibit controlled T2DM in response to metformin/SU combination therapy. In a mixed cohort composed of patients of European and African descent (African American), Rotroff et al. [25] showed that carriers of the CT and TT genotypes of rs2162145 may have a better response to metformin in comparison to carriers of the CC genotype. The study further demonstrated that carriers of the C allele of rs57081354 may have a decreased response to metformin [25]. With regards to rs254271, the authors demonstrated that Caucasian carriers of the CG and CC genotypes may have decreased response to metformin as compared to patients with genotype GG. Of note, this SNP was monomorphic among African Americans [25]. Literature suggest that African Americans are admixed in their African components of ancestry, with the majority contributions being from West and West-Central Africa. As such, the genetic architecture of African Americans is distinct from that of Africans [35]. Additionally, present day South Africans exhibit extensive genomic diversity in comparison to other populations groups [36]. On these grounds, it is possible for rs254271 to demonstrate different expression patterns among different groups of African origin. Genetic diversity may also be the reason for the disparities observed in the direction of association of the minor allele among people of European ancestry and South Africa Nguni people. Additional investigations conducted in a more diverse South African cohort are required to confirm the clinical impact of rs254271, rs2162145 and rs57081354 and further explore their potential as predictors of glycaemic response to anti-diabetic drugs.

In addition, the G allele and AG genotype of rs3792269 (*CAPN10*) were significantly associated with controlled T2DM in response to metformin/SU combination therapy in our cohort of South African Nguni (Swati and Zulu). The CAPN10 gene encodes calcium-dependent intracellular protease that is important in calcium-regulated signalling pathways [21]. The variant rs3792269 of *CAPN10* was previously associated with the preventive effect of metformin on the development of T2DM in subjects with pre-diabetic dysglycaemia [37]. This effect was observed among Caucasian carriers of the G allele who reside in Slovakia. However, the preventative effect of this SNP on the development of T2DM is yet to be established among people of African origin. In addition to preventing T2DM, it was demonstrated that European carriers of the minor allele G had a smaller probability of achieving HbA1c < 7% and they had a smaller reduction in HbA1c during the first six months of metformin treatment [21]. The differences observed in both studies could be explained by several factors. For instance, our study sampled patients who were on combination therapy, while Tkáč et al. [21] investigated patients on metformin monotherapy. Additionally, our study population of South African Nguni is different from the Central European Caucasian patients that were used in the reference study. While the direction of association of the genotypes and minor allele differed from previous findings, this SNP is proving to be of relevance in anti-diabetic treatment response among patients with T2DM.

In the present study, the GA and GG genotype of rs889299 were associated with uncontrolled T2DM in response to metformin/SU combination therapy. The A allele of the variant was previously associated with oedema in diabetic patients treated with Farglitazar and glibenclamide among Caucasians who reside in the United Kingdom (UK) [30]. The SNP rs889299 occur on the intronic region of *SCNN1B*, a gene that is responsible for providing instructions for the construction of the beta-subunit of ENac [30]. The activity of ENac is regulated by ATP-binding cassette protein such as the K channel-associated sulfonylurea receptor [31]. The K channel-associated sulfonylurea receptor is responsible for maintain energy balance within a living cell [14,38]. Sulphonylureas binds specific sites of this receptor, thereby blocking the inflow of K^+^ and stimulating the diffusion of Ca^+^ into the cytosol. This activity leads to the contraction of the filaments of actomyosin responsible for the exocytosis of insulin granules, which is; therefore, promptly secreted in large amounts [14,38]. While metformin has no effect on *SCNN1B*, its mechanism of action complements that of SUs by improving insulin sensitivity [11]. On these grounds, rs889299 can be used as a predictor for SU monotherapy or metformin/SU combination therapy. There is currently no record of the effect of rs889299 on glycaemic response to metformin and SU monotherapy or combination therapy. To the best of our knowledge, this is the first study to explore the effect of this polymorphism on glycaemic response to metformin/SU combination therapy in a population of African origin.

In the current study, we investigated genetic interactions between rs5708135 (*NBEA*), rs2162145 (*CPA6*), rs36056065 (*SLC22A1*), rs622342 (*SLC22A1*) and rs889299 (*SCNN1B*) and their effect on metformin/SU combination response. An interaction between rs2162145 and rs889299 was observed. Furthermore, the combination of TT (rs2162145) and GG (rs889299) as well as GA (rs889299) and TT (rs2162145) were prominently detected among uncontrolled patients. The GA (rs889299) and TT (rs2162145) combination was implicated in uncontrolled T2DM. Of note, the TT genotype of rs2162145 was associated with better response to metformin. The effect of this SNP may depend on the presence of rs889299, suggesting that both SNPs may synergistically influence glycaemic response to metformin/SU combination therapy among South African Nguni patients. The importance of gene-gene or SNP-SNP interactions is gaining recognition in the field of pharmacogenomics [39]. Epistatic interactions between rs594709 and rs2289669 in metformin efficacy among Chinese patients with T2D were reported by Xiao et al. (2016). Furthermore, Naja et al. [29] reported interactions between rs622342 (*SLC22A1*) and *CYP2C9*2* and *CYP2C9*3* associated with reduced levels of HbA1c in response to metformin/SU combination therapy among Lebanese patients. Only a few studies have explored this phenomenon with regards to anti-diabetic drugs; however, the importance of epistasis in anti-diabetic therapy is clearly identifiable. These findings have laid a foundation for the investigation of the complex interactions among genetic, and epigenetic factors that influence glycaemic response in metformin/ SU combination therapy among T2DM patients.

## 5. Limitations

Few limitations of the study cannot be ignored. The cross-sectional design does not allow for causal relationship to be established. Wide confidence interval and the high CVC score for the MDR model of interaction observed in the relationship between the SNPs and the glycaemic control is due largely to the small sample size. The absence of rs2683511 (TT), rs3792269 (GG) and rs72552763 (del/del) is noted. These genotypes were needed to better assess interaction effects that exist between these SNPs and glycaemic response to metformin/SU combination therapy. This is a health facility-based study with strict selection criteria (participants should have been initiated on combination therapy of metformin and SUs for at least a year at the time of the study). Thus, men were under-represented in the study due to their low utilisation of health facilities in the region. Low utilisation of health facilities by men has been reported extensively across South Africa. The proportion of men who utilise the healthcare system in the Eastern Cape Province ranged from 28.30% to 32.16%, as demonstrated by Adeniyi et al. [40] and Owolabi et al. [41]. Motala et al. [42], Adebolu et al. [43] and Olowe et al. [44] reported utilisation rates ranging between 20.48% and 30.00% in the KwaZulu Natal province. In the Western Cape, Erasmus et al. [45] and Peer et al. [46] reported rates ranging from 19.36 to 35.66%. Future studies should specifically target men and other ethnic populations at the community level in order to gain better understanding of the associations between SNPs on glycaemic response to metformin/SU combination therapy. Notwithstanding of these limitations, this study provides new insights into pharmacogenomics of metformin/SUs in South African adults with T2DM. In addition, this study has opened doors for pharmacogenomic studies in the ethnically-diverse population of South Africa.

## 6. Conclusions

This study reports the association of rs2162145 (CC), rs889299 (GA and GG) and *SLC22A1* rs622342 (CC) and rs57081354 (C) with uncontrolled T2DM in response to metformin/SU combination therapy in South Africa. The study also reports an association of rs254271 (C) and rs3792269 (G allele and genotype AG) with controlled T2DM. Furthermore, the study established an interaction between rs889299 and rs2162145 that is implicated in metformin/SU treatment outcome in an indigenous South African population. Further, pharmacogenomics and functional investigations should be conducted in a bigger South African cohort to confirm the effects of these genetic variants on metformin/SU combination therapy and provide more powerful evidence for their use as predictors of anti-diabetic treatment response.

## Figures and Tables

**Figure 1 jpm-11-00104-f001:**
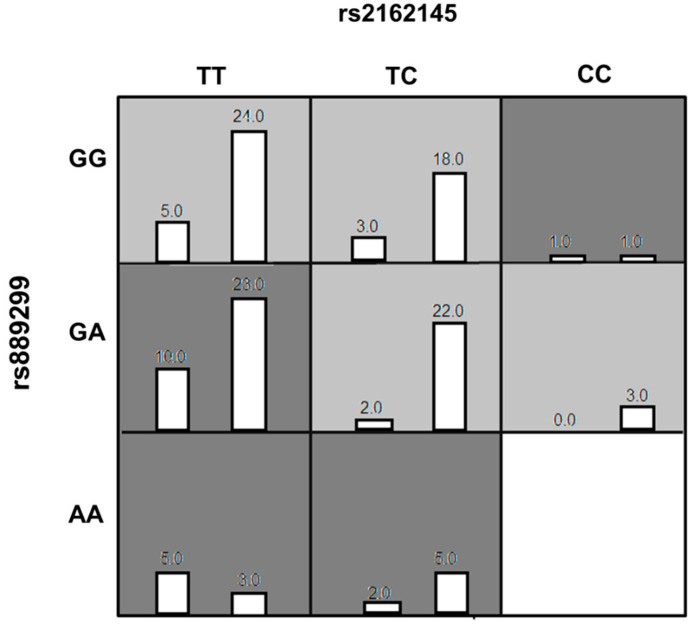
The best Multi-factor dimensionality reduction (MDR) model of interaction among rs2162145 and rs889299. The distributions of Controlled (left bars) and Uncontrolled (right bars) are illustrated for each combination of genotypes. Each cell represents genotype combinations. Dark grey cells represent genotype combinations implicated in uncontrolled type 2 diabetes mellitus (T2DM) in response to metformin/SU treatment. light grey cell represent genotype combinations implicated in controlled T2DM in response to metformin/SU combination therapy. White cells represent missing data. MDR= Multi-factor dimensionality reduction.

**Figure 2 jpm-11-00104-f002:**
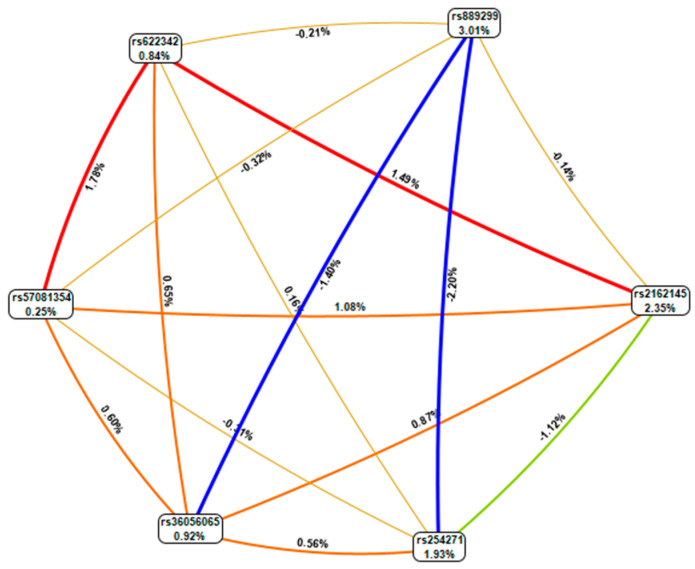
MDR combined attribute network showing all possible interactions between single nucleotide polymorphisms (SNPs). Each colour represents a possible interaction. Figures and line width indicate the strength of the interaction. Figures < 1 and thin lines represent weak interactions. The strongest interactions are represented by figures ≥1 and thick lines. The image was generate using open-source MDR software package version 3.0.2.

**Table 1 jpm-11-00104-t001:** General characteristics of the study cohort (BMI=body mass index; T2DM=type 2 diabetes mellitus).

Variable	Total (*n*; %)	Male (*n*; %)	Female (*n*, %)
	128 (100%)	18 (14.06)	110 (85.93)
**Ethnicity**			
Swati	22 (17.19)	04 (22.22)	18 (16.36)
Zulu	106 (82.81)	14 (77.78)	92 (83.64)
**Age**			
18–25 years	02 (1.56)	01 (5.55)	01 (0.91)
26–35 years	02 (1.56)	0 (0.00)	02 (1.81)
36–45 years	15 (11.72)	02 (1.56)	13 (11.81)
46–55 years	31 (24.22)	05 (27.77)	26 (23.63)
56–65 years	46 (35.93)	05 (27.77)	41 (37.27)
≥65 year	32 (25.00)	05 (27.77)	27 (24.54)
**BMI**			
<18.5 kg/m^2^	01 (0.78)	0 (0.00)	01 (0.91)
18.5–24.9 kg/m^2^	13 (10.16)	04 (22.22)	09 (8.18)
25.0–29.9 kg/m^2^	26 (20.31)	06 (4.69)	20 (18.18)
≥30 kg/m^2^	88 (68.75)	08 (4.44)	80 (72.72)
**T2DM treatment outcome**			
HbA1C ≤ 7%	29 (22.66)	06 (33.33)	45 (40.90)
HbA1C > 7%	99 (77.34)	12 (66.67)	65 (59.09)
**Duration of Diagnosis**			
<5 years	94 (73.44)	13 (72.22)	81 (73.64)
≥5 years	34 (26.56)	05 (27.78)	29 (26.36)
**Physical Activity**			
Active	50 (39.06)	08 (44.44)	42 (38.18)
Inactive	78 (60.93)	10 (55.56)	68 (61.82)

**Table 2 jpm-11-00104-t002:** SNP information and Hardy–Weinberg *p*-values for each SNP in the study cohort.

SNP	Gene	Chromosomal Location	Location	Amino Acid Change	Drug	HWE *p*-Value
rs2162145	*CPA6*	8:67747912	Intergenic	T > C	Metformin	0.771
rs2683511	*Sp1*	12:53410706	Intron	C > T	Metformin	0.670
rs3792269	*CAPN10*	2:240592062	Synonymous	A > G	Metformin	0.313
rs254271	*PRPF31*	19:54127382	Intron	G > C	Metformin	0.134
rs57081354	*NBEA*	13:35202457	Intron	C > T	Metformin	0.955
rs36056065	*SLC22A1*		Intron	GTAAGTTG > del	Metformin	0.002
rs622342	*SLC22A1*	6:160151834	Intron	A > C	Metformin	0.021
rs72552763	*SLC22A1*	6:160139849	Inframe Deletion	GAT > del	Metformin	0.522
rs889299	*SCNN1B*	16:23370593	Intron	A > G	Glibenclamide	0.504

Abbreviations: HWE = Hardy–Weinberg equilibrium; SNP = single nucleotide polymorphism

**Table 3 jpm-11-00104-t003:** Association of SNPs with glycaemic response to metformin/SU combination therapy.

dbSNP	Unadjusted Odds Ratios (95% CI)	*p*-Value	Adjusted Odds Ratios (95% CI)	*p*-Value	Bonferroni Adjusted *p*-Values
All					
**rs2162145**					
Genotypes					
TT	1		1		
CT	2.21 (0.45–10.71)	0.323	6.67 (0.99–46.22)	0.051	
CC	4.95 (0.84–29.00)	0.076	14.86 (1.71–129.04)	0.014	0.0035
Alleles					
T	1		1		
C	0.84 (0.42–1.65)	0.615	0.77 (0.37–1.61)	0.500	
**rs254271**					
Genotypes					
GG	1		1		
GC	0.29 (0.35–2.44)	0.257	0.10 (0.01–1.28)	0.077	
CC	0.27 (0.03–2.36)	0.738	0.13 (0.01–1.16)	0.119	
Alleles					
G	1		1		
C	1.34 (0.55–3.24)	0.513	0.20 (0.07–0.62)	0.005	0.00125
**rs2683511**					
Genotypes					
CC	1		1		
CT	1.25 (0.47–3.35)	0.648	1.61 (0.48–5.38)	0.433	
TT	-		-		
Alleles					
C	1		1		
T	1.08 (0.58–2.02)	0.791	1.09 (0.40–3.00)	0.856	
**rs36056065**					
Genotypes					
GG	1		1		
G/del	0.82 (0.20–3.23)	0.777	1.47 (0.28–7.51)	0.642	
del/del	0.63 (0.23–1.68)	0.358	0.60 (0.18–1.98)	0.405	
Alleles					
G	1		1		
del	0.83 (0.28–2.43)	0.733	1.15 (0.58–2.26)	0.682	
**rs3792269**					
Genotypes					
AA	1		1		
AG	0.29 (0.06–1.33)	0.113	0.15 (0.02–0.85)	0.033	0.00825
GG	-		-		
Alleles					
A	1		1		
G	0.33 (0.12–0.88)	0.027	0.87 (0.24–3.14)	0.835	
**rs5708135**					
Genotypes					
CC	1		1		
TC	0.84 (0.08–8.06)	0.885	1.37 (0.11–15.08)	0.800	
TT	0.85 (0.08–8.49)	0.890	1.83 (0.14–22.87)	0.637	
Alleles					
T	1		1		
C	2.15 (1.14–4.04)	0.017	0.33 (0.10–1.20)	0.056	
**rs622342**					
Genotypes					
AA	1		1		
CA	1.83 (0.58–5.71)	0.297	2.48 (0.51–12.05)	0.258	
CC	4.65 (1.08–19.86)	0.038	5.84 (1.00–34.07)	0.049	
Alleles					
A	1		1		
C	0.83 (0.22–3.08)	0.784	1.78 (0.89–3.52)	0.098	
**rs72552763**					
Genotype					
GG	1		1		
G/del	0.28 (0.03–2.31)	0.240	0.11 (0.01–1.56)	1.105	
del/del	-		-		
Allele					
G	1		1		
del	1.47 (0.76–2.84)	0.784	1.06 (0.20–5.61)	0.943	
**rs889299**					
Genotypes					
AA	1		1		
GG	3.55 (1.11–11.33)	0.032	7.91 (1.67–37.27)	0.009	0.00225
GA	2.66 (0.87–8.10)	0.084	5.27 (1.19–23.19)	0.028	0.007
Alleles					
A	1		1		
G	1.47 (0.76–2.84)	0.249	1.02 (0.48–2.15)	0.946	0.105

Abbreviations: CI = Confidence interval.

**Table 4 jpm-11-00104-t004:** Interaction models among the rs2162145, rs889299 and rs622342 in T2DM patients.

Interaction Models	Training Score	Testing Score	CVC	*p*-Value
*CPA6* rs2162145	0.6174	0.5099	7/10	0.0107
*CPA6* rs2162145 and SCNN1B rs889299	0.6749	0.4385	5/10	0.0004
*CPA6 rs2162145*, *SLC22A1* rs622342 and *SCNN1B* rs889299	0.7357	0.4742	6/10	0.0001

Abbreviations: CVC = Cross-validation consistency.

## Data Availability

All the study materials and data are available from the corresponding author, upon reasonable request.

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
