# Peer review of "Single Nucleotide Polymorphisms Associated with Metformin and Sulphonylureas’ Glycaemic Response among South African Adults with Type 2 Diabetes Mellitus"

_jpm, 2021, doi:10.3390/jpm11020104_

Round 1

Reviewer 1 Report

Charity Masilela * , Brendon Pearce , Joven Jebio Ongole , Oladele VINCENT ADENIYI , Mongi Benjeddou. Single Nucleotide Polymorphisms Associated with Metformin and Sulphonylureas’ Glycaemic Response among South African Adults with Type 2 Diabetes Mellitus

I have major corrections and comments to the authors:

The value of the study is not great, since the researchers conducted their work on a small sample. Sample needs to be increased. Moreover the authors write that «about 85.93% (n=110) of the study 25 participants were female and 77.34% (n= 99) had uncontrolled T2DM (HbA1c > 7%)». Why then were men included in the group? If such inclusion has been carried out, it is necessary to statistically prove the possibility of this association.

The abstract states that «In this study, we reported the association 30 of CPA6 rs2162145, SCNN1B rs889299, SLC22A1 rs622342 and NBEA rs57081354, PRPF31 rs254271 31 and CAPN10 rs3792269 with variable glycaemic response to metformin and SU combination 32 therapy». But no information is given in Results for the genes SLC22A1 rs622342 and NBEA rs57081354, PRPF31 rs254271 31 and CAPN10 rs3792269.

It is necessary not only to rewrite (and add) the entire results section, but also to recalculate the data again. In Section 3.2, there is practically no description of the results of this paragraph (p. 8). Bonferroni correction is not applied correctly (Table 3).

The discussion is written in an interesting way, but since the results are incorrect, it loses all meaning

The conclusion must be rewritten (after the work is completed) in accordance with the requirements for this part of article.

Please to use genetic nomenclature for alleles, genotypes.

Author Response

1.The value of the study is not great, since the researchers conducted their work on a small sample. Sample needs to be increased. Moreover the authors write that “about 85.93% (n=110) of the study participants were female and 77.34% (n= 99) had uncontrolled T2DM (HbA1c > 7%)”. Why then were men included in the group? If such inclusion has been carried out, it is necessary to statistically prove the possibility of this association.

Answer: The comment on the under-representation of men in the study sample is acknowledged. Given that this study was conducted in the health facilities in South Africa, our study participants reflect the overall health facility utilisation by the residents, which is predominantly females. Our study therefore opens door for larger studies at the community level across the ethnically-diverse population of the country.

  1. The abstract states that “In this study, we reported the association of CPA6 rs2162145, SCNN1B rs889299, SLC22A1 rs622342 and NBEA rs57081354, PRPF31 rs254271 31 and CAPN10 rs3792269 with variable glycaemic response to metformin and SU combination 32 therapy”. But no information is given in Results for the genes SLC22A1 rs622342 and NBEA rs57081354, PRPF31 rs254271 31 and CAPN10 rs3792269.

Answer: Thanks for the comment. We have modified the results and conclusion. 

  1. It is necessary not only to rewrite (and add) the entire results section, but also to recalculate the data again. In Section 3.2, there is practically no description of the results of this paragraph (p. 8). Bonferroni correction is not applied correctly (Table 3).

Answer: We agree with the reviewer and have corrected the error in the calculation.

  1. The discussion is written in an interesting way, but since the results are incorrect, it loses all meaning.

Answer: We have corrected the errors in our calculations, and all significant genotypes and alleles retained their significance. Thus, no changes were made in the discussion.

  1. The conclusion must be rewritten (after the work is completed) in accordance with the requirements for this part of article.

Answer: After adjusting our calculations, the significance did not shift. Therefore, our conclusion remains the same.

  1. Please to use genetic nomenclature for alleles, genotypes.

Answer: Thank you for the comment.

Reviewer 2 Report

Congratulations, your study is interesting and it was done very appropriately. However, it would have been appropriate to include more individuals of the Swati group and of course more males. Overall, your study demonstrates the importance to recognize the genetic difference in the treatment of T2DM.

This study was done very properly. However, here are my comments:

  1. In addition of shortage of male participants (14.06%), the majority of the participants are mostly over 65 years old (60.93%)
  2. Since the study involved individuals of Swati and Zulu groups, were there any difference of the presence of the genotypes and alleles responsible for uncontrolled T2DM in the two groups?
  3. It would be important to identify the genotypes and/or alleles or genotype combinations present in the individuals with controlled T2DM.
  4. Line 21, the abbreviation SNP should follow “Single Nucleotide Polymorphism”

Author Response

Reviewer 2:
It was done very appropriately. However, it would have been appropriate to include more individuals of the Swati group and of course more males. Overall, your study demonstrates the importance to recognize the genetic difference in the treatment of T2DM.
Here are my comments:
1. In addition to shortage of male participants (14.06%), the majority of the participants are mostly over 65 years old (60.93%)

Answer: Thanks for this insightful observation on our study participants – fewer males and Swati population group. It should be pointed out that our study participants are representative of individuals attending the health facilities in the region. Fewer men compared to women use health facilities. Notwithstanding the limitations, our study will definitely open doors for a larger community studies, involving more men and different ethnic groups in the country.

  1. Since the study involved individuals of Swati and Zulu groups, were there any difference of the presence of the genotypes and alleles responsible for uncontrolled T2DM in the two groups?

Answer: The number of Swati participants was very low. Therefore, it was not possible to make a fair comparison between the two ethnic groups.

  1. It would be important to identify the genotypes and/or alleles or genotype combinations present in the individuals with controlled T2DM.

Answer: Our MDR model highlights the most prominent genotype combinations in the controlled and uncontrolled groups (Figure 1).

  1. Line 21, the abbreviation SNP should follow “Single Nucleotide Polymorphism”

Answer: Thank you for the comment. The abbreviation has been corrected.

Round 2

Reviewer 1 Report

To my regret, I cannot accept the answer to either the first or second question. The calculations were performed incorrectly. The results are as poorly described as before. Remark # 6 ignored.

Author Response

Reviewer 1:
To my regret, I cannot accept the answer to either the first or second question. The calculations were performed incorrectly. The results are as poorly described as before. Remark # 6 ignored

Response: The Hardy-Weinberg Equilibrium (HWE), genotype and allele frequencies were calculated using a dedicated population genetic analysis software. Logistic and multivariate regression, and the corrective measures which accompany these analyses were calculated using both the MedCalc and SPSS packages for validity and quality checking. The authors believe that the statistical analyses and software used are sufficient for reporting data of this type. 

However, we welcome the suggestion of the reviewer on how best to re-analyse this data in order to further improve the results.

Previous comments from reviewer 1:

1.The value of the study is not great, since the researchers conducted their work on a small sample. Sample needs to be increased. Moreover, the authors write that “about 85.93% (n=110) of the study participants were female and 77.34% (n= 99) had uncontrolled T2DM (HbA1c > 7%)”. Why then were men included in the group? If such inclusion has been carried out, it is necessary to statistically prove the possibility of this association.

Answer: The comment of the reviewer on the fewer number of men in the study is acknowledged. Our study selected participants who had received diagnosis of DM and treatment with metformin and/or sulphonylureas for at least a year in the study sites. The study participants were limited by the strict selection criteria. As such, the number of men meeting these criteria were further limited due to the poor hospital utilisation by men in the region. Many studies across South Africa reported a low range of health utilisation by men between 19.36 - 35.66%: Motala et al., 2008 (20.48%); Erasmus et al., 2012 (19.36%); Peer et al., 2012 (35.66%); Adebolu et al., 2014 (30.00%); Adeniyi et al., 2016 (28.30%); Olowe et al., 2017 (21.55%); Owolabi et al., 2017 (32.16%). While a national survey involving a representative sample of health facilities across the country will ensure large proportion of all ethnic groups and gender, this is not feasible due to the resources available for this project. The authors undoubtedly believe that this study has opened doors for future studies among South African population on genetic association with treatment outcomes in individuals with diabetes.

  1. The abstract states that “In this study, we reported the association of CPA6 rs2162145, SCNN1B rs889299, SLC22A1 rs622342 and NBEA rs57081354, PRPF31 rs254271 31 and CAPN10 rs3792269 with variable glycaemic response to metformin and SU combination 32 therapy”. But no information is given in the Results for the genes SLC22A1 rs622342 and NBEA rs57081354, PRPF31 rs254271 31 and CAPN10 rs3792269.

Answer: Thanks for the comment. We have modified the results and conclusion. 

  1. Please to use genetic nomenclature for alleles, genotypes. Ignored by the authors.

Answer: Thank you for the comment. The authors chose to represent the alleles and genotypes in a similar pattern in accordance with previous authors who have recently published in JPM (Jha et al., 2020; AL-Eitan et al., 2020; Naja et al., 2020; Marquez Pete et al., 2020 and Lighezan et al., 2020).

Reviewer 2:

Comment: Congratulations, your study is interesting and it was done very appropriately. However, it would have been appropriate to include more individuals of the Swati group and of course more males. Overall, your study demonstrates the importance to recognize the genetic difference in the treatment of T2DM.

Response: Thank you for your comments. Indeed, the number of men was limited by the strict selection criteria (participants must have been diagnosed and initiated on treatment with Metformin and/or sulphonylureas for at least one year at the time of the study). Notwithstanding the limitations, our study has opened doors for bigger studies on the association of genetic factors with glycaemic control among South Africans with diabetes.